# Safety Profile of Homologous and Heterologous Booster COVID-19 Vaccines in Physicians in Quito-Ecuador: A Cross-Sectional Study

**DOI:** 10.3390/vaccines11030676

**Published:** 2023-03-16

**Authors:** Nancy Flores-Lastra, Josue Rivadeneira-Dueñas, Luis Fuenmayor-González, Glenda Guayasamín-Tipanta, Michelle Jácome-García, Tamara Otzen, Carlos Manterola

**Affiliations:** 1Zero Biomedical Research, Quito 170103, Ecuador; nancy.flores@zerobiomedicalresearch.com (N.F.-L.); lefuenmayor@uce.edu.ec (L.F.-G.);; 2Doctorado en Ciencias Médicas, Universidad de La Frontera, Temuco 4811230, Chilecarlos.manterola@ufrontera.cl (C.M.); 3Núcleo Milenio de Sociomedicina, Santiago 7560908, Chile; 4Facultad de Ciencias Médicas, Universidad Central del Ecuador, Quito 170136, Ecuador

**Keywords:** COVID-19 vaccines, BNT162 vaccine, ChAdOx1 nCoV-19, side effects and adverse reactions, booster immunizations, COVID-19, SARS-CoV-2

## Abstract

More than 600 healthcare workers died due to COVID-19 infection until January 2022 in Ecuador. Even though the COVID-19 vaccines are safe, local and systemic reactions were reported among physicians. This study aims to analyze the adverse events of COVID-19 with an emphasis on comparing the homologous and heterologous booster doses in physicians that received three approved vaccines in Ecuador. An electronic survey was performed in Quito, Ecuador, directed at physicians who were vaccinated with the three doses of COVID-19 vaccines. A total of 210 participants were analyzed after administering any dose of the vaccines. At least one AE was identified in 60.0% (126/210) of the sample after the first dose, 52.40% (110/210) after the second dose, and 75.2% (158/210) after the booster dose. The most frequent AEs were localized pain, myalgia, headache, and fever. At least one drug was used in 44.3% of the population after the first dose, 37.1% after the second dose, and 63.8% in the booster dose. Heterologous booster produces more AEs compared with homologous booster (80.1% vs. 53.8%), and 77.3% of participants reported that interfered with daily activities. Similar studies agree that reactogenicity occurs mainly with heterologous vaccination compared to homologous vaccination. This situation affected physicians’ performance in daily activities and led them to use medication for the symptoms. In the future, it is recommended to perform cohort studies, where adverse events that are associated with vaccine boosters in the general population can be analyzed longitudinally, thus improving the level of evidence of the results.

## 1. Introduction

The COVID-19 pandemic has produced 568,773,510 cases and 6,381,643 deaths worldwide [1]. In Ecuador, 946,487 people were diagnosed with COVID-19 [1]; the most affected province is Pichincha [2], and Quito, the capital of Ecuador, is the city with the highest number of confirmed cases [3].

In a historic context, the first COVID-19 vaccines arrived in Ecuador in January 2021 and were destinated for first responders [4], and four months later, in May 2021, the Nation Vaccination Plan included vulnerable groups such as the elderly, people with disabilities, and people with high occupational risk [5]. By July 2021, the population older than 16 got their vaccination [6], and in September 2021, nine million inhabitants got their full vaccination [7]. The booster dose application started on December 2021 with homologous and primarily heterologous regimens in those that were fully vaccinated 6 months prior [8].

Healthcare workers (HCWs) are defined as paid or unpaid persons engaged in actions whose primary intent is to enhance health [9]. Among the first line HCWs, physicians are one of the groups that are most affected by COVID-19 infections and the one with the highest mortality [10], for instance, more than 600 physicians died due to COVID-19 infection up to January 2022 in Ecuador [11]. For this reason, the World Health Organization (WHO) established that physicians were a priority in the administration of vaccines including the booster dose [12].

A booster dose is the response of the Public Health Systems to the emergence of new SARS-CoV-2 variants [13] and the decay of neutralizing antibodies in people that are fully vaccinated after 3–6 months [14,15]. The National Vaccination Campaign achieved full vaccination of 83.84% of the population and 39.98% got a booster dose [16]. The Ministry of Health (MoH) managed the administration of four COVID-19 vaccines: the mRNA Pfizer/BioNTech-*Comirnaty*, the non-replicating viral vector Oxford/AstraZeneca-*Vaxzevria*, and CanSino-*Convidencia*, and the inactivated virus vaccine Sinovac-*CoronaVac*.

The 56.39% of the total booster doses were primarily administered with Oxford/AstraZeneca-*Vaxzevria*, 34.78% with Pfizer/BioNTech-*Comirnaty*, 2.09% with CanSino-*Convidencia*, and 6.72% with Sinovac-*CoronaVac* [16].

Even though the COVID-19 vaccines are safe [17], a few local and systemic reactions were reported after the first and second doses of vaccination [18] and after the application of a heterologous booster dose among HCWs [19]. Regarding the homologous versus heterologous booster, the latter demonstrated a slightly higher rate of adverse events in HCW [20].

In Ecuador, the presence of adverse events associated with booster vaccination has not been studied. Due to the low vaccination rate related mainly to the fear and anxiety generated by the development of severe adverse events [21], it is important to provide local results to evaluate whether heterologous vaccination increases events [20], with the objective of reducing fear in the population and to provide physicians with another tool to stimulate vaccination.

This study aims to analyze the adverse events of COVID-19 with an emphasis on comparing the homologous and heterologous booster doses in physicians that received three approved vaccines in Ecuador.

## 2. Materials and Methods

This article was written following the STROBE initiative for the reporting of observational studies [22].

### 2.1. Study Design and Ethical Conditions

The present cross-sectional study was approved by the Independent Ethics Committee SOLCA-NUCLEO QUITO with the number OBS.21.202 and during its development, all the norms of Good Clinical Practice and Helsinki principles [23] were complied to maintain the participant’s safety and assure the quality of data that were obtained.

### 2.2. Settings, Data Source, and Participants

An open and anonymous electronic survey (27 questions and 6 different screens) was performed in the city of Quito, Ecuador, directed to the medical volunteer participants who were vaccinated with the 3 doses of the following vaccines against COVID-19: Oxford/AstraZeneca-*Vaxzevria*, Pfizer/BioNTech-*Comirnaty*, or Sinovac-*CoronaVac* (greater than 7 days from the third dose). The instrument was published on different online media and social networks using platforms followed by physicians to delimit the sample; obtaining data from February 15 to April 01 of 2022 through the Google formulary application after individual authorization through the acceptance of electronic Informed Consent. To deal with the risk of duplicated responses, the email address was requested and if duplicated files were identified then the oldest one was eliminated.

The validity of the instrument was determined through content validation based on the criteria of three experts (1 family physician, 1 Master in Public Health, and 1 general practitioner), after which the instrument was applied in a pilot study to 25 participants, who approved the understanding and clarity of the content. After this, a Cronbach’s Alpha of 0.89 was obtained for the instrument used, proving its reliability.

### 2.3. Measurement of Variables

*Outcome measures*: The results were obtained through the different responses to the questions that were asked during the development of the survey, keeping the development of any adverse event, the use of 1 or more medications, and the affectation of daily activities 7 days after any dose of vaccination against COVID-19 as dependent variables.

The outcome adverse events was evaluated through the closed question “Did you present any subsequent adverse event in the 7 days following the administration of the COVID-19 vaccine?”, in each dose administered (3 doses) dichotomizing the answer in Yes/No; in the case of an affirmative answer, the type of adverse event was requested through the following non-exclusive question “If yes. Which of the following adverse events occurred during the first 7 days post-vaccination?”, awaiting confirmation or denial of the following adverse events: localized pain, localized edema, localized erythema, localized pruritus, thermal elevation, diarrhea, nausea, vomiting, myalgia, arthralgia, headache, and the development of other types of adverse events (obtained from the U.S Food and Drug Administration (FDA) [24]).

The use of medication was requested through the closed question “Was any medication administered for these side effects?”, at each dose obtaining Yes/No answers; in case of an affirmative answer, it was requested to describe which medications were used through the use of the excluding question: Paracetamol, Ibuprofen, Diclofenac, Etoricoxib, and other medications. This qualitative variable was subclassified as the number of drugs that were used during the 7 days following each dose of the COVID-19 vaccine (None, 1 drug, 2 drugs, 3 or more drugs).

The affectation of daily activities during the 7 days after the administration of each dose of the COVID-19 vaccine was classified as an ordinal qualitative variable and assessed through a Likert-type scale with 4 options requested through the following question “Indicate the relationship between your daily activities and the adverse events present on a scale of 1 to 4”, obtaining, as a result, the options: (1) Does not interfere with your daily activities and no treatment is administered, (2) interferes with daily activities and/or required pharmacological treatment, (3) impedes the performance of daily activities and required treatment; and (4) was hospitalized.

*Individual measures*: The participants were asked for sociodemographic information related to their date of birth (for age calculation), sex (male, female), and place of work; this last variable was classified as public, private, or no work. Dates of vaccination and the name of the vaccine applied were also required for each dose; these variables were classified according to the number of doses (first dose, second dose, third dose) and administered vaccine (Oxford/AstraZeneca-*Vaxzevria*, Pfizer/BioNTech-*Comirnaty*, or Sinovac-*CoronaVac*). The third dose was subclassified concerning the type of vaccine into homologous (defined as administration of the same vaccine in all three doses) and heterologous (placement of a different vaccine in any dose) and is described in Table A1 in the Appendix A.

### 2.4. Biases

As the study requires retrospective data, the risk of memory bias is latent, and it was managed by placing the download link of the vaccination certificate of the MoH of Ecuador and the request based on mixed questions that counted as options for different adverse events that help to remember the events that occurred.

### 2.5. Statistical Analysis

To calculate the sample size, the finite universe formula was used with a sampling error of 5%, a confidence interval of 95%, and an expected proportion of 82.1% for a population of 11,730 physicians of Quito, Ecuador, with a loss of 10%, a sample of 250 participants was obtained. A non-probabilistic snowball sampling was used.

In the case of missing data, the mean of each variable was placed to avoid loss of information. The descriptive analysis for qualitative variables was expressed with frequency tables; for qualitative variables, summary measures such as the mean, median, standard deviation, and interquartile range were used and presented according to the type of distribution of each variable. For the bivariate analysis, the association between variables was assessed with a parametric test (chi^2^ and Student’s *t*-tests) and nonparametric test (Mann–Whitney U test), and risk measures (odds ratio) with their respective 95% confidence intervals, using the presence or absence of adverse events as the dependent variable. Statistical analysis was performed with the Statistical Package for the Social Sciences version 26 program^®^ (SPSS; IBM Corp., Armonk, NY, USA).

## 3. Results

A total of 225 participants gave their consent to participate in the study, obtaining a recruitment and completion rate of 100%; 15 research subjects were excluded, 13 for being non-medical professionals, and 2 for receiving another type of vaccine. A total of 210 participants were analyzed with 630 vaccines administered divided into 3 doses (Figure 1). A total of 65.7% (138/210) were women; the participants presented an average age of 32 years (RIQ: 11.25) and for the 630 doses that were administered, 47.3% were Pfizer/BioNTech-*Comirnaty*, the 39.0% Oxford/AstraZeneca-*Vaxzevria*, and 13.7% Sinovac-*CoronaVac*). Table 1 describes the characteristics of the participants; no missing data were identified during the analysis.

### 3.1. Doses Administered

It has been identified that 62.5% (394/630) of the doses that were administered presented at least one adverse event during the 7 days after the administration of any dose of the vaccine, with a frequency of 60.0% (126/210) after the first dose, 52.40% (110/210) and in the second dose, and 75.2% (158/210) after the third dose, this difference is statistically significant (*p*-value < 0.001). The most frequent adverse event that was reported was localized pain in 49.5% (312/630) followed by myalgia in 28.3% (178/630), headache in 24.9% (157/630), and thermal elevation with a frequency of 21.6% (136/630). Concerning the use of medications, 48.4% (305/630) of the participants reported having used at least one medication for the management of adverse events, 44.3% (93/210) after the first dose, 37.1% (78/210) after the second dose, and 63.8% (134/210) in the third dose applied. Of the participants who reported at least one adverse event, 44.7% (176/391) reported that their daily activities were affected and/or required pharmacological treatment. The characteristics of each dose are described in Table 2.

### 3.2. Vaccine Administered

Regarding the vaccine administered, it becomes evident that 74.8% (184/246) of the participants that were given Oxford/AstraZeneca-*Vaxzevria* presented adverse events, with a mean of 3.4 (SD = 2.0) adverse events. In the group of participants that were administered with Pfizer/BioNTech-*Comirnaty*, a frequency of 59.1% (176/298) with a mean of 2.4 (SD = 1.6) adverse events were identified, and in the Sinovac-*CoronaVac* group, 39.5% (34/86) presented at least one adverse event with a mean of 1.8 (SD = 1.2) this difference is statistically significant (*p*-value < 0.001). Concerning the use of medication, 63.0% (155/246) of the Oxford/AstraZeneca-*Vaxzevria* group reported using a medication, compared to 42.3% (126/298) of the Pfizer/BioNTech-*Comirnaty* group, and 27.9% (24/86) of the participants that were vaccinated with Sinovac-*CoronaVac*, this difference is statistically significant (*p*-value < 0.001). The characteristics of adverse events, the use of medication, and the relationship with daily activities are described in Table 3. In Appendix A, Table A2 provides descriptive characteristics of adverse events with the doses and the vaccine.

### 3.3. Group Vaccine

In the group of participants with heterologous reinforcement, it is evident that 80.1% (137/171) presented at least one adverse event, compared to 53.8% (21/39) of the homologous group, this difference is statistically significant (*p*-value < 0.001); they also report that 70.2% (120/171) report the use of medication for the control of adverse events in the heterologous group compared to 35.9% (14/39) of the homologous group, this difference is statistically significant (*p*-value < 0.001). The characteristics of adverse events, medication use, and their relationship with daily activities are described in Table 4.

### 3.4. Factors Related to the Occurrence of Adverse Events

In the bivariate analysis, it was identified that 62.5% (394/630) of the participants developed some adverse event, where being female (OR: 1.76 (1.25–2.46)), receiving the third dose (OR: 2.36 (1.64–3.42)), the Oxford/AstraZeneca vaccine (OR: 2.45 (1.79–3.49)), and a heterologous vaccine (OR: 4.20 (2.02–8.73)) increased the probability of presenting an adverse event. Table 5 describes these results.

## 4. Discussion

During the first year of the pandemic, scientists worldwide developed safe and effective vaccines to reduce hospitalization and deaths associated with COVID-19. However, due to the limited availability of vaccine doses, vaccination was prioritized for HCWs and was the best strategy to control deaths and diseases in this population [25]. It is essential to describe the characteristics of the local and systemic AEs presented by the HCW and to analyze their relationship with the degree of affectation in the performance of daily activities after full or booster immunization and thus improve vaccination plans in the future.

Our study shows a high prevalence of local and systemic adverse events (62.5%) in medical personnel after 7 days of administration of the approved vaccines in Ecuador, describing a higher frequency after administration of the third dose (75.2%), compared to the first and second doses (60.0% and 52.4%, respectively). Of these adverse events, the great majority did not interfere or slightly interfered with daily activities, showing that they were mild to moderate. Despite this, 48.4% of the participants administered some medication to control the side effects. Regarding the most frequent side effects, localized pain at the injection site (local event), myalgia, and headache (systemic events) were reported. Likewise, it was identified that being a woman, receiving an mRNA vaccine, Oxford/AstraZeneca-Vaxzevria, the third dose of any vaccine, and the heterologous vaccine as a booster increased the risk of presenting adverse events.

Naito et al. [26] conducted a study with HCWs in Japan with a population that received Pfizer/BioNTech-Comirnaty and Moderna vaccines. They reported similar frequencies of adverse reactions during the first eight days after the second and third vaccination. They also found that the incidence of adverse reactions was significantly higher in heterologous vaccination compared to homologous vaccination [26], which is consistent with our results and several others that have been reported [27,28,29].

In a German study performed by Nachtigall et al. [30], in health workers, the most reported local adverse event during the first eight days at any vaccination dose, was pain at the injection site (50.2%), followed by malaise (30.1%). Homologous vaccination produced fewer adverse events (49.4%) than heterologous vaccination (81.4%) [30].

In Thailand, a prospective study was performed in HCWs, where the most common systemic adverse events after the Sinovac-CoronaVac/Oxford/AstraZeneca-Vaxzevria booster dose included myalgia, headache, and fever (37%, 34%, and 34%). The AE prevalence was significantly lower for individuals in the Oxford/AstraZeneca-Vaxzevria group after the CoronaVac booster. The most common symptoms that were observed in this group were headache, myalgia, and fatigue (13%, 11%, and 9%, respectively), and the local reaction was 14%, in comparison with 17% of the other group [31], similar our results showed previously.

Concerning the vaccine type, Oxford/AstraZeneca-Vaxzevria vaccine showed the highest risk of adverse events (OR 2.45, CI 95% 1.73–3.49, *p* < 0.001) with similar results in several other studies [32,33]. On the other hand, a systematic review and meta-analysis performed by Kouhpayeh and Ansari reported that inactivated vaccines such as Sinovac-CoronaVac have no relationship with systemic adverse events (RR 1.13, CI 95% 0.79–1.61) [34] as in our study where Sinovac-CoronaVac showed the least risk for AE (OR 0.33, 95% CI 0.20–0.53, *p* < 0.001).

Besides the heterologous regimen during the third dose application, the highest probability of AEs with statistical significance was presented in females. These findings coincide with the results of Alqahtani et al., where female gender was a significant risk factor for developing AEs (OR: 1.69 CI 95% 1.17–2.44) [32]. On the other hand, Granados-Villalpando et al. analyzed AEs in the Mexican population, and even though the female gender was numerically a risk factor, the results were not statistically significant (OR: 1.69 CI 95% 1.17–2.44) [35].

Our study presented some limitations such as the low sample size, which could affect the power of the study. However, in post hoc examination, statistical power obtained 98%, with a low beta error. Moreover, the cohort consisted of physicians, and therefore, in terms of age distribution, most participants were of working age; this may constrain the generalizability of our results, especially for the elder age groups. In addition, the recall bias could also affect the reliability of the responses. Another limitation is the lack of multivariate analysis due to the lack of independence in the independent variables of the sample studied, and it is suggested that future studies take this type of analysis into account in order to reduce bias due to confounding variables. For these reasons, further analysis including larger samples of nurses, paramedics, and laboratory staff should be performed to generalize our results to the healthcare workers group.

## 5. Conclusions

This study presents the descriptive characteristics of local and systemic adverse events in medical personnel from Quito who were immunized with Oxford/AstraZeneca-Vaxzevria, Pfizer/BioNTech-Comirnaty, and Sinovac-CoronaVac during the National Vaccination Campaign in Ecuador. This study compares the symptoms that were described most frequently by the participants in the different doses and in the group that received homologous and heterologous vaccination. In addition, the relationship between vaccination and daily activities performance was analyzed, with the requirement of pharmacological treatment to alleviate the reported symptoms. People who received the Vaxzevria vaccine had more adverse events (74.8%) compared to those who received Comirnaty (59.1%) and CoronaVac (39.5%). The highest degree of adverse events occurred with heterologous vaccination compared to homologous immunization (80.1% vs. 53.8%), in line with other international results. All the vaccines that were used by medical personnel produced local and systemic adverse events from the first dose to the booster dose during the first seven days, while the booster dose presented a higher prevalence of adverse events. This situation led to the use of medication for the symptoms that presented and affected the performance of physicians’ daily activities during the pandemic.

## Figures and Tables

**Figure 1 vaccines-11-00676-f001:**
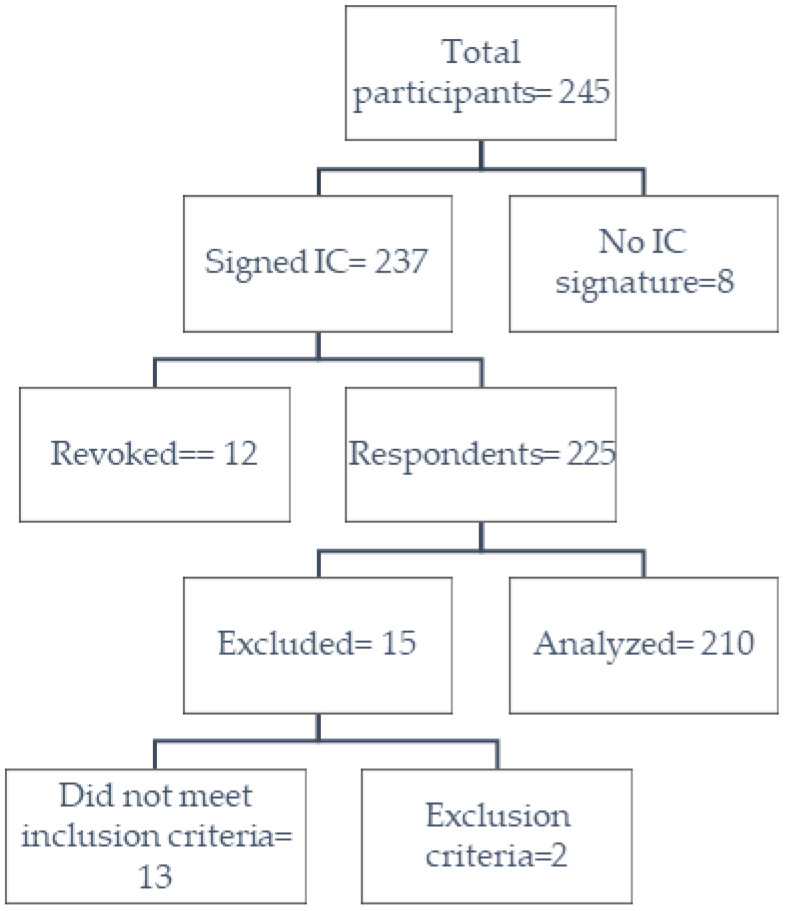
Summary of participants. IC: Informed consent.

**Table 1 vaccines-11-00676-t001:** Sociodemographic characteristics and vaccine that was administered to the participants.

		*n* (%)
General		210 (100.0)
Sex	Female	138 (65.7)
	Male	72 (34.3)
Age (Years)	Under 30 years of age	88 (41.9)
	30 to 39 years old	70 (33.3)
	40 to 49 years old	25 (11.9)
	50 to 59 years old	15 (7.1)
	Over 60 years old	12 (5.7)
Vaccine	Total	630 (100.0)
	Pfizer/BioNTech-*Comirnaty*	298 (47.3)
	Oxford/AstraZeneca-*Vaxzevria*	246 (39.0)
	Sinovac-*CoronaVac*	86 (13.7)

**Table 2 vaccines-11-00676-t002:** Descriptive characteristics of adverse events, medication, and relationship with daily activities.

	Total *n* (%)	First Dose *n* (%)	Second Dose *n* (%)	Third Dose *n* (%)	*p*-Value (*)
General	630 (100.0)	210 (33.3)	210 (33.3)	210 (33.4)	
Adverse Events	394 (62.5)	126 (60.0)	110 (52.4)	158 (75.2)	<0.001
Pain	312 (49.5)	107 (51.0)	91 (43.3)	114 (54.3)	0.071
Myalgia	178 (28.3)	45 (21.4)	36 (17.1)	97 (46.2)	<0.001
Headache	157 (24.9)	43 (20.5)	30 (14.3)	84 (40.0)	<0.001
Thermal elevation	136 (21.6)	30 (14.3)	19 (9.0)	87 (41.4)	<0.001
Arthralgia	111 (17.6)	29 (13.8)	17 (8.1)	65 (31.0)	<0.001
Edema	40 (6.3)	11 (5.2)	5 (2.4)	24 (11.4)	0.001
Pruritus	39 (6.2)	11 (5.2)	11 (5.2)	17 (8.1)	0.374
Erythema	36 (5.7)	14 (6.7)	6 (2.9)	16 (7.6)	0.084
Other types of adverse events	100 (15.9)	30 (3.6)	16 (7.3)	54 (25.8)	0.008
Use of medication	305 (48.4)	93 (44.3)	78 (37.1)	134 (63.8)	<0.001
1 drug	243 (79.7)	84 (90.3)	68 (87.2)	91 (67.9)	-
2 drugs	54 (17.7)	6 (6.5)	8 (10.3)	40 (29.9)	-
3 or more drugs	8 (2.6)	3 (3.2)	2 (2.6)	3 (2.2)	-
Daily Activities	394 (100.0)	126 (100.0)	110 (100.0)	158 (100.0)	-
Do not interfere with your daily activities and no treatment is administered	158 (40.1)	64 (50.8)	56 (50.9)	38 (24.1)	-
Interferes with daily activities and/or required pharmacological treatment.	176 (44.7)	55 (43.7)	46 (41.8)	75 (47.5)	-
Impedes the performance of daily activities and required treatment	57 (14.5)	7 (5.6)	8 (7.3)	42 (26.6)	-
Was hospitalized	3 (0.8)	0 (0.0)	0 (0.0)	86 (1.9)	-

*: Chi-square test.

**Table 3 vaccines-11-00676-t003:** Descriptive characteristics of adverse events, medication, and relationship to daily activities in relation to the vaccine.

	Oxford/AstraZeneca-*Vaxzevria* *n* (%)	Pfizer/BioNTech-*Comirnaty* *n* (%)	Sinovac-*CoronaVac* *n* (%)	*p*-Value (*)
General	246 (100.0)	298 (100.0)	86 (100.0)	
Adverse Events	184 (74.8)	176 (59.1)	34 (39.5)	<0.001
Pain	129 (52.4)	152 (51.0)	31 (36.0)	0.025
Myalgia	111 (45.1)	62 (20.8)	5 (5.8)	<0.001
Headache	94 (38.2)	53 (17.8)	10 (11.6)	<0.001
Thermal elevation	91 (37.0)	41 (13.8)	4 (4.7)	<0.001
Arthralgia	78 (31.7)	32 (10.7)	1 (1.2)	<0.001
Edema	23 (9.30)	13 (4.4)	4 (4.7)	0.047
Pruritus	18 (7.3)	18 (6.0)	3 (3.5)	0.443
Erythema	16 (6.5)	16 (5.4)	4 (4.7)	0.767
Other types of adverse events	62 (22.3)	38 (12.7)	0 (0.0)	0.006
Use of medication	155 (63.0)	126 (42.3)	24 (27.9)	<0.001
1 drug	111 (71.6)	110 (87.3)	22 (91.7)	-
2 drugs	40 (25.8)	12 (9.5)	2 (8.3)	-
3 or more drugs	4 (2.6)	4 (3.2)	0 (0.0)	-
Daily Activities	184 (100.0)	176 (100.0)	34 (100.0)	-
Do not interfere with your daily activities and no treatment is administered	54 (77.0)	79 (44.9)	25 (73.5)	-
Interferes with daily activities and/or required pharmacological treatment.	88 (79.0)	79 (44.9)	9 (26.5)	-
Impedes the performance of daily activities and required treatment	39 (18.0)	18 (10.2)	0 (0.0)	-
Was hospitalized	3 (1.6)	0 (0.0)	0 (0.0)	-

*: chi-square test.

**Table 4 vaccines-11-00676-t004:** Adverse events after vaccination medication, and relationship to daily activities: heterologous vs. homologous.

	Heterologous *n* (%)	Homologous *n* (%)	*p*-Value (*)
**General**	171 (100.0)	39 (100.0)	
Adverse Events	137 (80.1)	21 (53.8)	<0.001
Pain	97 (56.7)	17 (43.6)	0.156
Myalgia	89 (52.0)	8 (20.5)	<0.001
Thermal elevation	80 (46.8)	7 (17.9)	0.001
Headache	78 (45.6)	6 (15.4)	<0.001
Arthralgia	61 (35.7)	4 (10.3)	0.002
Edema	23 (13.5)	1 (2.6)	0.055
Pruritus	15 (8.80)	2 (5.1)	0.745
Erythema	14 (8.20)	2 (5.1)	0.742
Other types of adverse events	51 (29.9)	3 (7.7)	<0.001
Use of medication	120 (70.2)	14 (35.9)	<0.001
1 drug	82 (68.3)	9 (64.3)	-
2 drugs	35 (29.2)	5 (35.7)	-
3 or more drugs	3 (2.5)	0 (0.0)	-
Daily Activities	137 (100.0)	21 (100.0)	-
Do not interfere with your daily activities and no treatment is administered	28 (20.4)	10 (47.6)	-
Interferes with daily activities and/or required pharmacological treatment.	68 (49.6)	7 (33.3)	-
Impedes the performance of daily activities and required treatment	38 (27.7)	4 (19.0)	-
Was hospitalized	3 (2.2)	0 (0.0)	-

*: chi-square test.

**Table 5 vaccines-11-00676-t005:** Factors related to the development of adverse events.

		Total (*n*/%)	Adverse Event No (*n*/%)	Adverse Event Yes (*n*/%)	*p* Value (*)	OR (CI 95%)
**Overall**		630 (100.0)	236 (37.5)	394 (62.5)	-	-
Sex						
Female		414 (65.7)	136 (57.6)	278 (70.6)	0.001	1.76 (1.25–2.46)
Male (+)		216 (34.3)	100 (42.4)	116 (29.4)		
Age						
Over or equal 32 years old		522 (82.9)	193 (81.8)	329 (83.5)	0.579	1.12 (0.73–1.72)
Under to 32 years of age (+)		108 (17.1)	43 (18.2)	65 (16.5)		
Doses administered (+)						
First Dose	Yes	210 (33.3)	84 (35.6)	126 (32.0)	0.353	0.85 (0.60–1.19)
	No (+)	420 (66.7)	152 (64.4)	268 (68.0)		
Second Dose	Yes	210 (33.3)	100 (42.4)	110 (27.9)	<0.001	0.52 (0.37–0.74)
	No (+)	420 (66.7)	136 (57.6)	284 (72.1)		
Third Dose	Yes	210 (33.3)	52 (22.0)	158 (40.1)	<0.001	2.36 (1.64–3.42)
	No (+)	420 (66.7)	184 (78.0)	236 (59.9)		
Vaccine						
Pfizer/BioNTech-*Comirnaty*	Yes	298 (47.3)	122 (51.7)	176 (44.7)	0.087	0.75 (0.55–1.04)
	No (+)	332 (52.7)	114 (48.3)	218 (55.3)		
Oxford/AstraZeneca-*Vaxzevria*	Yes	246 (39.0)	62 (26.3)	184 (46.7)	<0.001	2.45 (1.73–3.49)
	No (+)	384 (61.0)	174 (73.7)	210 (53.3)		
Sinovac-*CoronaVac*	Yes	86 (13.7)	52 (22.0)	34 (8.6)	<0.001	0.33 (0.20–0.53)
	No (+)	544 (86.3)	184 (78.0)	360 (91.4)		
Group Vaccine						
Heterologous		171 (81.4)	51 (29.8)	120 (70.2)	<0.001	4.20 (2.02–8.73)
Homologous (+)		39 (18.6)	25 (64.1)	14 (35.9)		

CI: Confidence interval; OR: Odds Ratio. *: Chi-square test; +: Reference variable; dependent variable: Development of any adverse event; Independent variables: Sex, Age, Number of doses, Vaccine and group Vaccine.

## Data Availability

The data that are presented in this study are available on request from the corresponding author. The data are not publicly available due to maintaining privacy data of the participants such as e-mail addresses.

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
