# Peer review of "Safety Profile of Homologous and Heterologous Booster COVID-19 Vaccines in Physicians in Quito-Ecuador: A Cross-Sectional Study"

_vaccines, 2023, doi:10.3390/vaccines11030676_

Round 1
Reviewer 1 Report
Hello, The paper is good read on the first look. However, I have few suggestion for improving the manuscript.
1. Tha abstract may be modified a bit to include basic aim and objectives of the work. Also, future directions must be mentioned.
2. The introduction may be improved by adding proper rationale of the work.
3. The discussion section must be improved by including proper supportive studies to compare and contrast the findings.
Author Response
Response to Reviewer 1 Comments
Point 1: Tha abstract may be modified a bit to include basic aim and objectives of the work. Also, future directions must be mentioned.
Response 1: We accept the comment. The abstract was improved including a brief description of the objectives and future directions.
Point 2: The introduction may be improved by adding proper rationale of the work.
Response 2: We accept the comment. The reviewer’s suggestions have improved the introduction. We include a timeline of Ecuador’s COVID-19 vaccination and justification as well as a better rationale for the work.
Point 3: The discussion section must be improved by including proper supportive studies to compare and contrast the findings.
Response 3: We accept the comment. The discussion has been improved with the reviewer's recommendations. We add new comparatives between related studies and our results in terms of the risk factors of female gender, and heterologous regimen.
Kind regards.
Nancy Flores-Lastra, MS
Josué Rivadeneira-Dueñas, MD
Luis Fuenmayor-González, MD
Reviewer 2 Report
The article has important biases in the methods, in the results, and therefore in the whole study. Evaluating adverse events after a vaccination through a survey is too risky and unscientific. it is true that the subjects to whom the survey was submitted are doctors, however it is not possible to know if they were all adequately informed about the potential side effects. Another essential bias is the fact that the side effect is presumed but it is not certain that it is definitely attributable to the vaccine, it can be a simple coincidence. Finally, the study offers no contribution, the alleged side effects are in line with the indications in the drug label.
Author Response
Response to Reviewer 2 Comments
Point 1: The article has important biases in the methods, in the results, and therefore in the whole study. Evaluating adverse events after a vaccination through a survey is too risky and unscientific. it is true that the subjects to whom the survey was submitted are doctors, however it is not possible to know if they were all adequately informed about the potential side effects. Another essential bias is the fact that the side effect is presumed but it is not certain that it is definitely attributable to the vaccine, it can be a simple coincidence. Finally, the study offers no contribution, the alleged side effects are in line with the indications in the drug label.
Response 1: We kindly received the reviewer’s comments. Nevertheless, the study performed was a replication of several studies that assessed adverse events through electronic surveys. The limitations and potential biases are clearly reported, and the methods, results, and discussion sections were improved with reviewers´ comments.
Kind regards.
Nancy Flores-Lastra, MS
Josué Rivadeneira-Dueñas, MD
Luis Fuenmayor-González, MD
Reviewer 3 Report
The emphasis of the study is to compare the homologous and heterologous vaccine regimes but this objective is not indicated by the title of the manuscript.
There is a need to provide the history of COVID-19 infection in the country along with timeline of COVID-19 vaccination, in general. Afterwards, the focus can be reverted to healthcare professionals.
The information on validation and reliability analysis of the survey is not provided in the methods section.
Why did the author not perform the sample size estimation? The sample of 210 is very low to have a good power of the study.
Based on the three vaccines, there are many regimes for example Astra+astra+astra, astra+astra+pfizer, astra+pfizer+astra, Pfizer+astra+astra, pfizer+pfizer+astra....................and so on.... (It is not clear whether the authors consider which combination as homologous or heterologous). It would be clearer if the authors could provide a detailed and clear definition of these two regimes.
The discussion section is very concise. The authors have determined the factors associated with side effects but all these factors are not discussed in the discussion section. It is important to describe that why the side effects were common in certain demographics such as female, age 32 years. Why are the side effects more common after the third dose, regardless of heterologous or homologous regime. Why heterologous regime causes more side effects than homologous vaccines?
The logistic model has several confounders. For example, the third dose has more side effects and has higher odds of side effects. However, these odds are impacted by the heterologous regime too. I think that the authors should consider the adjusting of the models
The limitation section is very brief. First and foremost, the limitation of this study is sample size, then recall bias, etc.
The rationale of the study is not clear. There are some studies on the side effects in the country then why this study is aimed and what literature gap is covered by this study?
Table 2-4 are descriptives. There should be inferential statistics to compare the groups in these tables. Side effects can be compared between first, second and third dose by chi-square.
Author Response
Response to Reviewer 3 Comments
Point 1: The emphasis of the study is to compare the homologous and heterologous vaccine regimes but this objective is not indicated by the title of the manuscript.
Response 1: We accept the comment. The title has been updated.
Point 2: There is a need to provide the history of COVID-19 infection in the country along with timeline of COVID-19 vaccination, in general. Afterwards, the focus can be reverted to healthcare professionals.
Response 2: We accept the comment. The introduction has been improved with the reviewer’s suggestions, including a timeline of Ecuador’s COVID-19 vaccination.
Point 3: The information on validation and reliability analysis of the survey is not provided in the methods section.
Response 3: We accept the comment. In the methods section, we included the validation analysis of the survey through experts’ opinions, fiability tests, and a pilot test.
Point 4: Why did the author not perform the sample size estimation? The sample of 210 is very low to have a good power of the study.
Response 4: The sample size estimation was performed and is evidenced in the methods section (2.5 statistical analysis). However, since the expected sample of 225 participants was not obtained, the post hoc statistical power was calculated, obtaining a statistical power of 98%.
Point 5: Based on the three vaccines, there are many regimes for example Astra+astra+astra, astra+astra+pfizer, astra+pfizer+astra, Pfizer+astra+astra, pfizer+pfizer+astra....................and so on.... (It is not clear whether the authors consider which combination as homologous or heterologous). It would be clearer if the authors could provide a detailed and clear definition of these two regimes.
Response 5: We accept the coment, Appedix A table A1 was added including the description of different regimes.
Point 6: The discussion section is very concise. The authors have determined the factors associated with side effects but all these factors are not discussed in the discussion section. It is important to describe that why the side effects were common in certain demographics such as female, age 32 years. Why are the side effects more common after the third dose, regardless of the heterologous or homologous regime. Why heterologous regime causes more side effects than homologous vaccines?
Response 6: We accept the comment. The discussion has been improved with the reviewer's recommendations (description of the risk factors of female gender, and heterologous regimen).
Point 7: The logistic model has several confounders. For example, the third dose has more side effects and has higher odds of side effects. However, these odds are impacted by the heterologous regime too. I think that the authors should consider the adjusting of the models
Response 7: In our study, logistic regression analysis was not performed due to the existence of confounding variables; a bivariate analysis was simply executed by association and magnitude tests. Likewise, multivariate analysis was not performed because the assumption of independence of the independent variables was not fulfilled. Therefore, the recommendation could not be accepted; however, this explanation is added to the limitations reported.
Point 8: The limitation section is very brief. First and foremost, the limitation of this study is sample size, then recall bias, etc.
Response 8: We accept the comment. A more significant paragraph on limitations was included at the end of the discussion.
Point 9: The rationale of the study is not clear. There are some studies on the side effects in the country then why this study is aimed and what literature gap is covered by this study?
Response 9: We understand the reviewer’s comment. The literature gap covered by our study was the description of adverse events in physicians comparing the homologous vs heterologous booster regimens (these definitions were described above). Two lines specifying this were added at the end of the introduction
Point 10: Table 2-4 are descriptives. There should be inferential statistics to compare the groups in these tables. Side effects can be compared between first, second and third dose by chi-square.
Response 10: We accept the comment. Inferential analysis is increased in the requested tables.
Kind regards.
Nancy Flores-Lastra, MS
Josué Rivadeneira-Dueñas, MD
Luis Fuenmayor-González, MD
Round 2
Reviewer 2 Report
I have made considerations and the authors have not made any significant changes to the original manuscript. I don't see any improvements so the paper should be rejected.
Author Response
Dear reviewer.
We really appreciate your suggestions and sincerely apologize because the changes we made do not meet your requirements, however, we want to emphasize the importance of developing this type of study in Latin American countries, such as Ecuador, where the socio-cultural characteristics differ from the rest of the countries and may generate different results.
Similarly, the use of electronic surveys to assess adverse events in vaccination has been a useful method in several published studies.
Best regards.
Reviewer 3 Report
I have no more concerns on this manuscript.
Author Response
Dear reviewer.
Thank you for the changes previously suggested. They contributed to our knowledge and the quality of the article presented.
Best regards